# The Glyoxalase System Is a Novel Cargo of Amniotic Fluid Stem-Cell-Derived Extracellular Vesicles

**DOI:** 10.3390/antiox11081524

**Published:** 2022-08-05

**Authors:** Rita Romani, Vincenzo Nicola Talesa, Cinzia Antognelli

**Affiliations:** Department of Medicine and Surgery, University of Perugia, L. Severi square, 06129 Perugia, Italy

**Keywords:** glyoxalases, methylglyoxal, MG-H1, extracellular vesicles, human amniotic fluid, stem cells, D-lactate, GSH

## Abstract

The glyoxalase system is a ubiquitous cellular metabolic pathway whose main physiological role is the removal of methylglyoxal (MG). MG, a glycolysis byproduct formed by the spontaneous degradation of triosephosphates glyceraldehyde-3-phosphate (GA3P) and dihydroxyacetonephosphate (DHAP), is an arginine-directed glycating agent and precursor of the major advanced glycation end product arginine-derived, hydroimidazolone (MG-H1). Extracellular vesicles (EVs) are a heterogeneous family of lipid-bilayer-vesicular structures released by virtually all living cells, involved in cell-to-cell communication, specifically by transporting biomolecules to recipient cells, driving distinct biological responses. Emerging evidence suggests that included in the EVs cargo there are different metabolic enzymes. Specifically, recent research has pointed out that EVs derived from human amniotic fluid stem cell (HASC-EVs) contain glycolytic pay-off phase enzymes, such as glyceraldehyde-3-phosphate dehydrogenase (GAPDH). Since GAPDH catalyzes the sixth step of glycolysis using as a substrate GA3P, from which MG spontaneously origins, we wanted to investigate whether MG-derived MG-H1, as well as glyoxalases, could be novel molecule cargo in these EVs. By using immunoassays and spectrophotometric methods, we found, for the first time ever, that HASC-EVs contain functional glyoxalases and MG-H1, pioneering research to novel and exciting roles of these eclectic proteins, bringing them to the limelight once more.

## 1. Introduction

Glyoxalases are highly conserved and ubiquitously expressed metabolic enzymes, responsible for the detoxification of methylglyoxal (MG), a spontaneous by-product of energy metabolism, into non-toxic D-lactic acid [1]. Glyoxalases, comprising glyoxalase 1 (Glo1) and glyoxalase 2 (Glo2), have been long considered part of the glyoxalase system, working sequentially to remove MG. In particular, Glo1 catalyzes the formation of S-D-lactoylglutathione (SLG) from the hemithioacetal formed non-enzymically from MG and reduced glutathione (GSH), while Glo2 catalyzes the hydrolysis of SLG to D-lactic acid and regenerates the GSH consumed in the Glo1-catalysed reaction [2]. Additionally, indeed, this seems to be true in several biological settings [1,3]; however, in other scenarios, emerging evidence suggests that both enzymes may play distinct, novel, and independent roles [4,5,6,7]. MG is mainly formed as a glycolysis byproduct by the spontaneous degradation of the triosephosphates, glyceraldehyde-3-phosphate (GA3P), and dihydroxyacetonephosphate (DHAP) and is a highly reactive cellular metabolite that glycates lysine and arginine residues to form post-translational modifications (PTM), known as advanced glycation end products (AGEs) [8]. The major MG-dependent arginine-derived AGE is the hydroimidazolone MG-H1 [1]. MG-derived dicarbonyl adducts exert complex pleiotropic effects, including modulation of protein biological activity [9] and stability [10,11]. In general, targeted proteins are differently vulnerable to MG-dependent glycation so that their functionality can be impaired [11] or enhanced [12,13]. Interestingly, modifications were recently observed on the active site residues of glycolytic enzymes that could alter their activity [8] and in pivotal stress-inducible proteins implicated in cellular recovery, including heat shock protein 27 (Hsp27), increasing their functionality [12,13]. Moreover, there is also evidence that some MG-derived AGEs are endowed with antioxidant properties [14]. Notably, MG can also nonenzymatically modify lipids and nucleic acids [15,16,17,18]. Finally, MG and MG-derived AGEs are associated with free radical production, the increased activity of prooxidant enzymes, the decrease in antioxidant activities, and mitochondrial dysfunction [7].

Extracellular vesicles (EVs) are nanometer-sized membranous vesicles secreted by cells, with crucial roles in physio-pathological processes [19]. EVs play an important role in intercellular communication by transferring a heterogeneous biological cargo made of proteins, including misfolding proteins, lipids, and nucleic acids [20,21].

A growing body of evidence suggests that EVs contain different metabolic enzymes [22,23,24,25,26].

Interestingly, it has been very recently pointed out that EVs derived from human amniotic fluid stem cell (HASC-EVs) contain glycolytic pay-off phase enzymes, including glyceraldehyde-3-phosphate dehydrogenase (GAPDH) [27]. Since GAPDH catalyzes the sixth step of glycolysis using as a substrate GA3P, from which MG spontaneously and inevitably originates, we wondered whether glyoxalases, that metabolize MG, could be molecule cargo in these EVs, and whether their presence was paralleled by MG-derived MG-H1, the major MG-dependent arginine-derived AGE is the hydroimidazolone MG-H1 [1].

Interestingly, we found, for the first time to our knowledge, that HASC-EVs contain functional glyoxalases and MG-H1, pioneering research to novel and exciting roles of these eclectic proteins.

## 2. Materials and Methods

### 2.1. Materials

All the reagents used in the present study were of analytical grade from Sigma-Aldrich (Milan, Italy), unless otherwise specified. The bicinchoninic acid (BCA) kit for protein quantification was from Thermo Fisher Scientific (Monza, Italy).

### 2.2. HASC Isolation and Culture

HASCs were previously obtained from human amniotic fluid of 16–17 weeks pregnant women (35–40 years) who underwent amniocentesis. Women had no gestational diabetes or other previous diseases and were homogeneous for age. The study was approved by the University of Perugia Bioethics Committee. Written informed consent was obtained from each woman. HASCs were isolated and cultured as previously described [26,27,28]. Briefly, fresh amniotic fluid (3–5 mL) was centrifuged and the cell pellet cultured in 18% Chang B/2% Chang C media (Irvine Scientific, Ireland, UK) for 6–7 days to allow adherent cells to form colonies. Cells with a predominantly fibroblast-like morphology and a colony shape similar to dermatoglyphics, appropriately characterized, represented HASCs and were cultured in MSCGM medium (Lonza, Gaithersburg, MD, USA). Since HASCs characterization in all the cases showed similarity in all the parameters considered for their characterization [26,27,28], only one HASC cell line was considered to recover EVs.

### 2.3. HASC-Derived EV Isolation and Culture

EV isolation was performed as previously described [26,27]. Briefly, serum-free conditioned medium (MSCBM Lonza, Milan Italy # PT-3238 with L-glutamine and gentamicin sulfate/amphotericin B) from 5 × 10^6^ HASCs was pooled 24 h post-culturing, and sequentially centrifuged: 300× *g* (10 min), 2000× *g* (20 min), and 10,000× *g* (45 min). The pellet was named HASC-P10 fraction. Then, the supernatant was further centrifuged at 100,000× *g* (60 min) in an Optima TLX ultracentrifuge with a 60Ti rotor (Beckman Coulter, Brea, CA, USA) and the pellet named HASC-P100 fraction. Both HASC-P10 and HASC-P100 pellets were washed at 100,000× *g* (60 min) with PBS containing 1% penicillin/steptomycin. The pellets were suspended at 1 mg/mL concentration with endotoxin-free PBS (Merck, Darmstadt, Germany) added with 1% penicillin/streptomycin to avoid potential microbial contamination and stored at −80 °C until further use. All samples were tested for their endotoxin level with Pierce™ LAL Chromogenic Endotoxin Quantitation Kit (Thermo Scientific, San Jose, CA, USA). All the samples used in the study had endotoxin levels below the detection limit.

### 2.4. HASC-Derived EV Characterization

Nanoparticles tracking analysis (NTA), Scanning Electron Microscopy (SEM), Transmission Electron Microscopy (TEM) analyses as well as specific EVs markers were performed to characterize HASC-derived EVs as previously described [27].

Notably, HASC-P10 and HASC-P100 fractions used in the present study represent aliquots from those used in Mezzasoma et al. [27]. Results on both HASC-EVs fraction characterization are reported in Mezzasoma et al. [27].

### 2.5. Western Blot Analysis

Samples were run on 4–15% SDS-PAGE and blotted onto a nitrocellulose membrane (iBlot Dry Blotting System, Thermo Fisher Scientific, Monza, Italy). Membranes were blocked in Roti-Block for 1 h at room temperature and subsequently incubated overnight at 4 °C with an appropriate dilution of the following primary Abs: anti-Glo1 pAb, E-AB-15072, diluted 1:1000; anti-GLO2 pAb, ab154108, diluted 1:1000; anti-β-actin mAb, sc-376421, diluted 1:1000; anti MG-H1 mAb, STA-011, dilution 1:1000; anti-RAGE (A-9) mAb, sc-365154, dilution 1:1000; Hsp27 pAb, PA1-017, dilution 1:500. After washing, membranes were incubated for 1 h at RT with the appropriate HRP-conjugated secondary Ab and visualized using ECL (Amersham Pharmacia, Milan, Italy). The primary Ab was then stripped by incubating membranes in stripping buffer (100 mM 2-ME, 2% SDS, and 62.5 mM Tris-HCl, pH 6.8) and re-probed with an Ab against an appropriate housekeeping protein as an internal loading control. Original blots are reported in Appendix A.

### 2.6. Glyoxalase-Specific Activity

Glo1 activity was studied as previously described [29]. Glo2 activity was measured as previously described [30]. Protein concentration was determined with the BCA kit, using bovine serum albumin as a standard.

### 2.7. MG-H1 Detection

MG-H1 levels were detected using an OxiSelect Methylglyoxal Competitive ELISA kit (DBA Italia, Milan, Italy) [31].

### 2.8. Immunoprecipitation

Immunoprecipitation (IP) was performed using a Dynabeads Protein G Immunoprecipitation Kit (Thermo Fisher Scientific, Monza, Italy) according to the manufacturer’s instructions and as reported previously [32].

### 2.9. Glutathione (GSH) Assay

GSH was measured by using the GSH assay kit (colorimetric) from Bio Vision Inc. (Milpitas, CA, USA) according to the manufacturer’s instructions as previously described [33].

### 2.10. D-Lactate Assay

D-lactate was measured by using the D-Lactate Assay Kit (Colorimetric) from Abcam (Cambridge, MA, USA), according to the manufacturer’s instructions.

### 2.11. Statistical Analysis

Unless otherwise stated, results are expressed as means ± SD from three independent experiments and evaluated by Student’s *t*-test. A *p* value less than 0.05 was considered significant.

## 3. Results

### 3.1. Glyoxalases Are Cargos of Both HASC-P10 and HASC-P100 EVs

Figure 1 shows Glo1 and Glo2 protein levels in lysed HASC-P10 and HASC-P100 EVs. As shown, both fractions contain Glo1 and Glo2 proteins (Figure 1a). Interestingly, both Glo1 and Glo2 were more expressed in HASC-P10 than HASC-P100 EVs, when normalized against β-actin (Figure 1b).

### 3.2. Glyoxalases Are Functional in Both HASC-P10 and HASC-P100 EVs

In order to investigate whether glyoxalases were functional in HASC-P10 and HASC-P100 EVs, the specific activity of both enzymes was measured by spectrophotometry. We found that both glyoxalases were active in HASC-P10 and HASC-P100 EVs (Figure 2). In line with the results from WB, both Glo1 and Glo2 specific activity was higher in HASC-P10 compared with HASC-P100 EVs. Exposure of HASC-P10 and HASC-P100 EVs to proteinase K (PK), which degrades proteins localized on the EV surface, confirmed the presence of glyoxalases inside EVs (data not shown).

### 3.3. HASC-P10 and HASC-P100 EVs Contain MG-H1

It has been recently shown that the HASC-EVs used in the present study contain, among other proteins, glyceraldehyde-3-phosphate dehydrogenase (GAPDH) [27], the glycolytic enzyme catalyzing the oxidation of GA3P, from which MG spontaneously and inevitably originates [1]. This observation, together with the finding that Glo1, the major scavenger of MG, was housed in HASC-EVs, prompted us to investigate whether HASC-EVs could contain also MG-derived MG-H1, the most abundant MG-dependent arginine-derived AGE [1]. Indeed, we found that both vesicles contained MG-H1, whose levels were higher in HASC-P10 than in HASC-P100 EVs (Figure 3a,b).

### 3.4. The Receptor for Advanced Glycation End Products (RAGE) Is not Present in HASC-P10 and HASC-P100 EVs

It is known that MG-H1 can act through the receptor for AGEs, RAGE [34]. By using an Ab specific for a region of the protein common to both full-length cell surface or soluble RAGE, we did not find RAGE expression in both HASC-P10 and HASC-P100 EVs (Figure 4).

### 3.5. Heat Shock Protein (Hsp)27 Is a MG-H1-Modified Protein in HASC-P10 and HASC-P100 EVs

Among the proteins preferentially modified by MG, there are those belonging to the Heat Shock protein (Hsp) family [35,36]. Since a qualitative proteomic analysis previously performed in the same HASC-P10 and HASC-P100 EVs that we used in the present study, identified, among others, Hsps, including Hsp27 [27], and one of the major bands in our MG-H1 WB sized approximately around 27 KDa (Figure 3a), we wanted to see whether this protein was indeed modified by MG also in EVs. As shown in Figure 5a, we found that Hsp27 was a MG-H1-modified protein. Densitometric analysis of MG-H1-Hsp27 levels relative to total Hsp27 in IP samples showed that MG-H1-Hsp27 levels were higher in P10 compared with their expression in P100 EVs (Figure 5b), paralleling the trend of MG-H1 levels (Figure 3a).

### 3.6. HASC-P10 and HASC-P100 EVs Contain GSH and D-lactate

To further characterize the glyoxalase pathway in HASC-EVs, we measured in both HASC-P10 and HASC-P100 the levels of GSH and D-lactate. We found that both HASC-EVs contained GSH and D-lactate and that their levels were higher in HASC-P10 than in HASC-P100 EVs (Figure 6).

## 4. Discussion

The glyoxalase pathway, composed of Glo1 and Glo2 with GSH as cofactor, plays a crucial role in the chemical detoxification of MG, a highly reactive agent, inevitably formed as a by-product of glycolysis during the conversion of triose phosphate isomers (GA3P and DHAP), in all living organisms. Accumulation of MG can lead to the formation of AGEs. In particular, MG is an arginine-directed glycating agent and precursor of the major AGE arginine-derived MG-H1. Since preventing increased intracellular MG and/or MG-H1 is mandatory for the viability of a cell, the glyoxalase pathway is a key mechanism in maintaining MG and/or MG-H1 physiological levels, with Glo1 being in the top 10% most abundant cytosolic proteins [37]. While MG can be released from healthy or injured cells, glyoxalases, unless released by damaged cells, are typically intracellular enzymes. In the present study, we found, for the first time to our knowledge, that glyoxalases and the whole system (MG-H1, GSH, D-lactate), are contained in HASC-EVs, thus, suggesting novel unexploited roles for these polyhedric proteins, not only in the context of amniotic fluid, where HASC-EVs were obtained from, but potentially also in broader contexts. Moreover, while we were completing our study, it has been published that circulating neuronal EVs from mild cognitive impairment and different stages of Alzheimer’s disease patients contain Glo1 [38].

In particular, our results show that both Glo1 and Glo2 proteins are enveloped in HASC-P10 and HASC-P100 EVs and are functional, as shown by their specific activity detection.

Moreover, both HASC EVs contain Glo1-dependent MG-derived MG-H1 and D-lactate, the latter specifically generating by Glo2-catalysed reaction, thus, further supporting Glo1 and Glo2 functionality. Curiously, Glo1 specific activity in HASC-P10 EVs was higher than in HASC-P100 EVs and positively correlated with Glo2 activity, MG-H1, and D-lactate levels. The correlation between Glo1 and Glo2 activity suggests that the two enzymes traditionally work in these EVs as part of the glyoxalase system, which is not always so [1,5]. The correlation between Glo1 activity and MG-derived MG-H1 probably indicates a major Glo1 functionality demand to modulate the higher intra-vesicle MG-derived MG-H1 levels and guarantee an adequate amount of this MG-AGE for the role it plays in these EVs, which is all to be explored. In other words, Glo1 seems to be adequately fitted to handle MG and, consequently, MG-H1 levels. Moreover, the hormetic potential of MG has been previously proposed in a specific ambit [39], which does not exclude that it may occur also in other contexts. On the other hand, we speculate that the presence of a higher amount of MG-H1 in the HASC-P10 EVs fraction could be related to the lower GAPDH expression, previously detected in the same EVs fraction used in this study [27]. In fact, GAPDH converts GA3P, from which MG spontaneously and inevitably originates [1], to 1,3-bisphosphoglycerate. For more, GA3P is the product of aldolase-catalyzed reaction, another glycolytic enzyme identified in the proteomic analysis of the HASC EVs used here [27]. Hence, in HASC-P10 EVs where GAPDH is less expressed, it is plausible that its substrate GA3P accumulates and spontaneously degrades to MG that, consequently, generates MG-H1. Neither Glo1 nor Glo2 transcript levels were detected in HASC EVs, which suggests that both enzymes are synthesized inside HASC and subsequently released through EVs.

Emerging evidence indicates the presence of mitochondrial components in EVs [40]. As known, Glo2 is also located in the mitochondrion [7], which might suggest a mitochondrial origin for EV-embedded Glo2. However, since no mitochondrial protein has been detected in our EVs [27], this hypothesis seems to be hardly plausible.

MG-derived AGEs, including MG-H1, exert complex pleiotropic effects, including control of protein functionality [9] and stability [10,11]. In general, targeted proteins are differently vulnerable to MG-dependent glycation so that their functionality can be impaired [11] or enhanced [12,13]. Interestingly, modifications were recently observed on the active site residues of pivotal stress-inducible proteins implicated in cellular recovery, including heat shock protein 27 (Hsp27), increasing their functionality [12,13]. Here, we found that a major MG-H1-modified protein in HASC-P10 and HASC-P100 EVs was just Hsp27, already identified by the qualitative proteomic analysis previously performed in the same HASC EVs [27]. Hsp27 is a multifaceted intracellular protein. Inter alia, it responds to stress and behaves as a chaperone [41]; it acts as an antioxidant to scavenge reactive oxygen species; it regulates actin polymerization during cellular stress conditions [42]. Recently, Hsp27 have been found to be exported by exosomes, playing a key role in cell-to-cell communication, signaling, immunity, and inflammation. Interestingly, Hsp27 also plays a pivotal role in the biogenesis of exosomes [40]. We speculate that MG-modified Hsp27 may be endowed with a major functionality in performing the above-mentioned biological roles, which remain to be demonstrated. Of course, additional MG-modified proteins must be identified to further explore alternative functions for MG-H1 in HASC EVs.

MG-H1 can signal by binding to RAGE [34]. Differently from other studies that detected the presence of RAGE on neuronal derived-exosomes [38,43], RAGE was not expressed in both HASC EVs, indicating that the function of these EVs, as well as the role of MG-H1, does not depend on this receptor.

At present, we do not know why the glyoxalase system is a cargo of HASC EVs. According to the results of the present explorative study, what we can more plausibly hypothesize is that the system is incorporated in HASC EVs to possibly ensure hormetic levels of MG and MG-H1, spontaneously originating inside these independent metabolic units via glycolysis [27], in order to maintain vesicle stability, potentially, through Hasp27 modification. However, of course, a multitude of other hypotheses could be put forward. Just to mention a few: can the system contribute to ATP production? ATP has been proved to be produced in the HASC EVs used in our investigations [27]. Or, can Glo2 substrate, SLG, be an energy source for EVs? It is known that the free energy liberated during the hydrolysis of the thioester bond of SDL is even greater than that liberated during hydrolysis of the phosphoanhydride bond of ATP [44]. Or can Glo2 be inside EVs to catalyze protein S-glutathionylation using SLG as a substrate [45] for whatever reason? Or can the system be involved in EVs biogenesis? Or can the system be a novel mediator in intercellular communication from donor to recipient cells? Remarkably, EVs have been shown to affect several biological processes, such as cell migration, angiogenesis, inflammation, tissue repair and regeneration through their paracrine cargo [46]. Hence, the glyoxalase system could be a novel previously undiscovered actor in all these biological responses. In addition, since EVs derived from stem cells may be envisioned as attractive medicinal therapeutic products for future cell-free paracrine treatment, also in consideration of their presumed low immunogenicity [46], the glyoxalase system could also play a role in this fascinating applicative ambit. Interestingly, since also autophagosomes can be a source of EVs, through the “secretory autophagy” pathway [47], and autophagy, as well as glyoxalases, participate to maintain cellular homeostasis and longevity [48,49], it is also fascinating to hypothesize that glyoxalase-containing EVs might be somehow involved in aging control.

Similarly, we do not know what the glyoxalase system is doing inside EVs from amniotic fluid-derived stem cells. It could be just one of the protagonists in the maintenance of EV structure, stability, energy demand, as speculated above, and/or a novel player in stem-cell-dependent tissue restoration and regeneration [50] during pregnancy or even parturition, as occurs for other cargos [51]. Or it could represent an additional diagnostic marker to check molecular transformations of the fetal status, as well as serving as a predictor of ongoing pregnancy and preterm labor, as described for other EV cargos, including AGEs [52,53,54].

At present we do not have the answers to all these questions, but we strongly believe that our pioneering study may give a positive input to start a novel research field involving the glyoxalase system.

## 5. Conclusions

In the present study, we found for the first time that the glyoxalase system is a novel cargo of amniotic fluid stem-cell-derived extracellular vesicles. Secretion into the extracellular milieu through vesicles suggests additional or novel exciting functions of this eclectic system in addition to their intracellular properties. We have just discovered the tip of a huge iceberg.

## Figures and Tables

**Figure 1 antioxidants-11-01524-f001:**
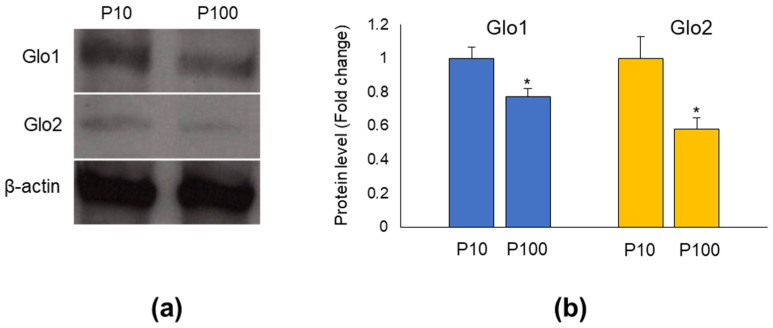
Glyoxalases are cargos of both HASC-P10 and HASC-P100 EVs. (**a**) Representative Western blot (WB) of Glo1 and Glo2; (**b**) quantitative histogram of the relative Glo1 and Glo2 protein expression levels. β-actin was used as internal loading control for WB normalization. The WB bands of Glo1 and Glo2 were quantified by densitometric analysis, normalized against β-actin, and expressed as relative protein level (fold change). The histograms indicate the mean ± SD of three independent experiments. * *p* < 0.05 vs. P10.

**Figure 2 antioxidants-11-01524-f002:**
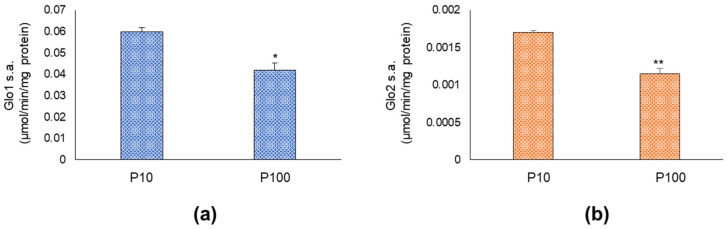
Glyoxalases are functional in both HASC-P10 and HASC-P100 EVs. (**a**) Glo1 and (**b**) Glo2 specific activity (s.a.), measured by specific spectrophotometric methods as described in Materials and Methods. The histogram indicates mean ± SD of three independent measurements, each tested in triplicate. * *p* < 0.05; ** *p* < 0.01.

**Figure 3 antioxidants-11-01524-f003:**
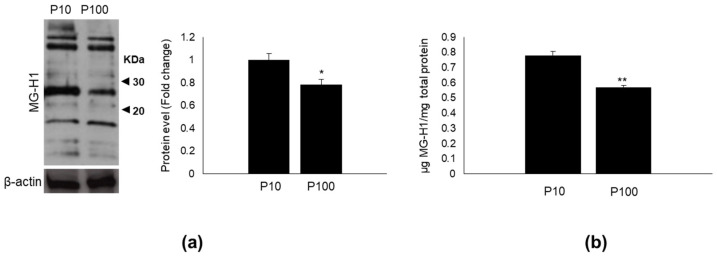
HASC-P10 and HASC-P100 EVs contain MG-H1. (**a**) Representative Western blot (WB) of MG-H1 and quantitative histogram of the relative MG-H1 protein expression levels. β-actin was used as internal loading control for WB normalization. The WB bands of MG-H1 were quantified by densitometric analysis (sum of multi-bands intensity), normalized against β-actin, and expressed as relative protein level (fold change). (**b**) Levels of MG-H1, measured by a specific ELISA kit. The histograms indicate the mean ± SD of two independent experiments. * *p* < 0.05 and ** *p* < 0.01 vs. P10.

**Figure 4 antioxidants-11-01524-f004:**
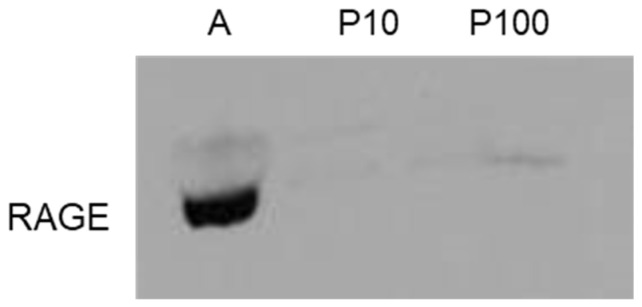
Protein level of the Receptor for Advanced Glycation End Products (RAGE) in HASC-P10 and HASC-P100 EVs, evaluated by Western blot. A: Positive control (human primary osteoblasts, PromoCell (Heidelberg, Germany) [34]).

**Figure 5 antioxidants-11-01524-f005:**
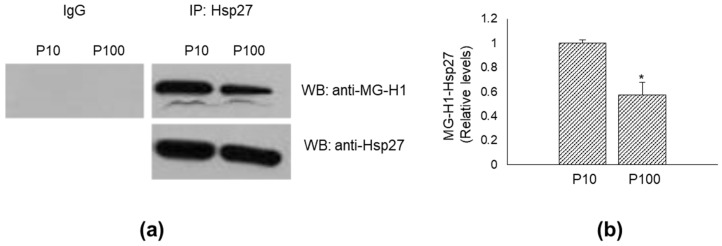
Heat shock protein (Hsp)27 is a MG-H1-modified protein in HASC-P10 and HASC-P100 EVs. (**a**) Lysates from HASC-P10 and HASC-P100 EVs were immunoprecipitated with protein G agarose-coupled anti-Hsp27 (IP: Hsp27) and subjected to Western blotting (WB) with anti-MG-H1 antibody. Blots were then stripped and re-probed with anti-HSP27 antibody (WB: anti-Hsp27) to ensure equal immunoprecipitation of HSP27 proteins. IgG was used as negative control for immunoprecipitation; (**b**) the histogram indicates mean ± SD of MG-H1-Hsp27 relative to total Hsp27 in IP samples, as quantified by densitometric analysis of WB bands. Normalized optical density values were expressed as relative protein level units. * *p* < 0.01 vs. HASC-P10.

**Figure 6 antioxidants-11-01524-f006:**
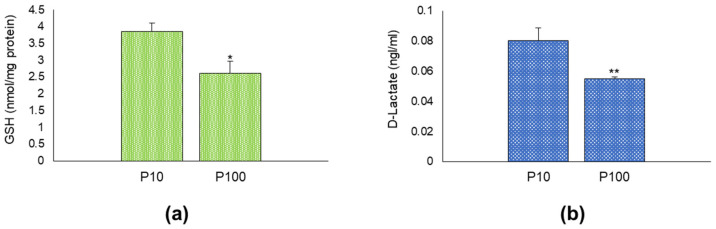
HASC-P10 and HASC-P100 EVs contain GSH and D-lactate. (**a**) Levels of GSH, measured by a specific GSH assay kit, as described in Materials and Methods; (**b**) levels of D-Lactate, measured by a specific D-lactate assay kit, as described in Materials and Methods. The histograms indicate the mean ± SD of two independent experiments. * *p* < 0.05 and ** *p* < 0.01 vs. P10.

## Data Availability

The data presented in this study are available in the article and Appendix A.

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
