# Peer review of "The Glyoxalase System Is a Novel Cargo of Amniotic Fluid Stem-Cell-Derived Extracellular Vesicles"

_antioxidants, 2022, doi:10.3390/antiox11081524_

Round 1
Reviewer 1 Report
In this Communication, authors provide, for the first time ever, a very enthusiastic and novel description of the glyoxalase system as a novel cargo of amniotic fluid stem cell-derived extracellular vesicles.
The manuscript is clearly written, nicely arranged and well referenced. References are updated. Results are methodologically validated, not just hypothesis, well presented and scientifically appropriate for a “communication”. Discussion is very intriguing, being centered on the several potential roles that the system could play in these vesicles and in cellular responses.
In parallel, another crucial strength of the study is its originality which opens new pathways of investigation aimed at improving knowledge on this metabolic pathway also in biological ambits we never expected. In conclusion, this is really a very interesting and important pioneering study that I’m sure will break new ground on glyoxalases and related metabolites.
English is fine.
I just would like to suggest authors to consider an additional interesting hypothesis they could further discuss, regarding the potential involvement of the vesicular glyoxalase system in aging-associated autophagy. As known, autophagy is a process in which cell macromolecules are engulfed in a double-membrane vesicle, the autophagosome, which could regulate different cell processes, such as aging, and we know that the role of glyoxalases in aging is well consolidated by now.
Author Response
Dear Reviewer,
thank you very much for providing helpful comments and suggestions aimed at improving the manuscript and strengthen the impact of our research work.
Detailed responses addressing point-by-point the issues raised in the individual comments are provided below.
"In this Communication, authors provide, for the first time ever, a very enthusiastic and novel description of the glyoxalase system as a novel cargo of amniotic fluid stem cell-derived extracellular vesicles. The manuscript is clearly written, nicely arranged and well referenced. References are updated. Results are methodologically validated, not just hypothesis, well presented and scientifically appropriate for a “communication”. Discussion is very intriguing, being centered on the several potential roles that the system could play in these vesicles and in cellular responses. In parallel, another crucial strength of the study is its originality which opens new pathways of investigation aimed at improving knowledge on this metabolic pathway also in biological ambits we never expected. In conclusion, this is really a very interesting and important pioneering study that I’m sure will break new ground on glyoxalases and related metabolites. English is fine."
Authors’ response: The authors thank the Reviewer very much for his/her appreciation on our study.
"I just would like to suggest authors to consider an additional interesting hypothesis they could further discuss, regarding the potential involvement of the vesicular glyoxalase system in aging-associated autophagy. As known, autophagy is a process in which cell macromolecules are engulfed in a double-membrane vesicle, the autophagosome, which could regulate different cell processes, such as aging, and we know that the role of glyoxalases in aging is well consolidated by now."
Authors’ response: The authors thank the Reviewer for his/her very interesting suggestion. We have now added a comment in the Discussion section according to what suggested (please, see pag. 9, lines 341-345).
Reviewer 2 Report
To date, there are many studies devoted to the proteomic analysis of extracellular vesicles, which have identified several thousand proteins contained in extracellular vesicles. However, based on these data, it is impossible to draw conclusions about the functional significance of these proteins. The presented work was analyzed the functional role of two enzymes, such as glyoxalase-I and glyoxalase-II, contained in two fractions of extracellular vesicles. Enzymatic system consisting of glyoxalase-I and glyoxalase-II essential for detoxification of methylglyoxal to d-lactate. The authors showed that the extracellular vesicles contain all the intermediates necessary for the functioning of this enzymatic system. The obtained results of the studies make a significant contribution to deciphering the puzzle of the functional activity of extracellular vesicles. In addition to the manuscript, there are several comments on the quality of the data presentation.
How many individual amniotic fluid donors were included in the study? Was there reproducibility of the results between donors in terms of the content of glyoxalase-I and glyoxalase-II in extracellular vesicles?
When studying extracellular vesicles, it is important to provide evidence that the isolated objects meet the criteria for extracellular vesicles. Although the methods describe that NTA and electron microscopy were used, data from these analyzes are not presented. It is also necessary to use surface markers to identify extracellular vesicles such as tetraspanins.
The Materials and Methods section should list all antibodies used in the appropriate dilutions.
Please, indicate the source of obtaining human primary osteoblasts.
What was the purpose of the analysis of extracellular vesicles for the content of endotoxins?
It is extremely interesting that the amount of enzymes (glyoxalase-I and glyoxalase-II) differs depending on the fraction of extracellular vesicles. It has recently been shown that ectosomes and exosomes contain different amounts of mitochondrial proteins and possibly elements of these organelles (doi: 10.3390/ijms23137408.). The authors believe that the differences in the concentrations of glyoxalase-I and glyoxalase-II can be explained by the localization of these enzymes in mitochondria within the HASC-P10 fraction? Perhaps this issue needs to be discussed.
Author Response
Dear Reviewer,
thank you very much for providing helpful comments and suggestions aimed at improving the manuscript and strengthen the impact of our research work.
Detailed responses addressing point-by-point the issues raised in the individual comments are provided below.
"To date, there are many studies devoted to the proteomic analysis of extracellular vesicles, which have identified several thousand proteins contained in extracellular vesicles. However, based on these data, it is impossible to draw conclusions about the functional significance of these proteins. The presented work was analyzed the functional role of two enzymes, such as glyoxalase-I and glyoxalase-II, contained in two fractions of extracellular vesicles. Enzymatic system consisting of glyoxalase-I and glyoxalase-II essential for detoxification of methylglyoxal to d-lactate. The authors showed that the extracellular vesicles contain all the intermediates necessary for the functioning of this enzymatic system. The obtained results of the studies make a significant contribution to deciphering the puzzle of the functional activity of extracellular vesicles. In addition to the manuscript, there are several comments on the quality of the data presentation."
Authors’ response: The authors thank the Reviewer for his/her appreciation on our study and comments/suggestions aimed at improving the manuscript and strengthen the impact of our research work.
"How many individual amniotic fluid donors were included in the study? Was there reproducibility of the results between donors in terms of the content of glyoxalase-I and glyoxalase-II in extracellular vesicles?"
Authors’ response: As already reported in M&M of the original version of our Communication (“2.4. HASC-derived EVs characterization…….Notably, HASC-P10 and HASC-P100 fractions used in the present study represent aliquots from those used in Mezzasoma et al. [27]…..”), in the present study, we used residual material of HASC-P10 and HASC-P100 fractions, recently used in the study by Mezzasoma et al., Ref. 27. The HASCs from which the P10 and P100 EVs were obtained came from human amniotic fluid collected from different pregnant women over time. These HASCs had already been previously characterized (doi: 10.1111/jcmm.12534 (Ref. 26), doi: 10.1039/c5mb00018a), and turned to be identical, stable for numerous passages (over 200) and did not undergo phenotypic changes during cryopreservation. Hence, since all the HASCs were identical over the other for all the parameters considered in their characterization, including the proteomic profile (doi: 10.1111/jcmm.12534 (Ref. 26), doi: 10.1039/c5mb00018a), only one HASC cell line was considered to recover EVs. A short informative paragraph has been now added in M&M at “2.2. HASCs isolation and culture”. Pease, see pag. 2, paragraph 2.2.
"When studying extracellular vesicles, it is important to provide evidence that the isolated objects meet the criteria for extracellular vesicles. Although the methods describe that NTA and electron microscopy were used, data from these analyzes are not presented. It is also necessary to use surface markers to identify extracellular vesicles such as tetraspanins."
Authors’ response: Again, in the present study, we used residual material of HASC-P10 and HASC-P100 fractions recently employed in the study by Mezzasoma et al. (Ref. 27), where results from NTA and electron microscopy as well as EVs markers had been already showed (Figure 1 of ref 27). For this reason, as stated in M&M (Results on both HASC-EVs fraction characterization are reported in Mezzasoma et al. [27].), these results have not been shown here, but for them to be observed, we have reminded to ref [27] and two additional papers (doi: 10.1111/jcmm.12534 (Ref. 26), doi: 10.1039/c5mb00018a).
"The Materials and Methods section should list all antibodies used in the appropriate dilutions."
Authors’ response: The used antibodies and relative dilution has been now added in M&M.
"Please, indicate the source of obtaining human primary osteoblasts."
Authors’ response: The source of human primary osteoblasts has been added. However, it was also reported in Ref. 33, cited in the Legend.
"What was the purpose of the analysis of extracellular vesicles for the content of endotoxins?"
Authors’ response: Analysis of endotoxins was performed in order to exclude potential bacterial contamination that might occur during EVs isolation that could have affected results. This was already explained in the original version of the manuscript, please see Paragraph 2.3 of M&M.
"It is extremely interesting that the amount of enzymes (glyoxalase-I and glyoxalase-II) differs depending on the fraction of extracellular vesicles. It has recently been shown that ectosomes and exosomes contain different amounts of mitochondrial proteins and possibly elements of these organelles (doi: 10.3390/ijms23137408.). The authors believe that the differences in the concentrations of glyoxalase-I and glyoxalase-II can be explained by the localization of these enzymes in mitochondria within the HASC-P10 fraction? Perhaps this issue needs to be discussed."
Authors’ response: The Reviewer’s consideration is indeed right. However, no mitochondrial markers were found in our EVs (please, see Figure 1 of Ref. 27, Mezzasoma et al.). By the way, only the differences in the concentrations of Glo2 could have been potentially explained by the localization of this enzyme in mitochondria, since Glo1 is not present in these organelles. Anyway, a comment in Discussion has been now made, starting from the very interesting article, doi: 10.3390/ijms23137408. Please, see pag. 8., lines 293-296.
Reviewer 3 Report
The MS of Romani R et al. „The glyoxalase system is a novel cargo of amniotic fluid stem 2 cell-derived extracellular vesicles“ presents novel and interesting data. It would be potentially interesting to readers and for the Journal, however, critical details are missing to support the conclusions and interpretation of findings in the present version. Authors provide very scarce information on the source of biological material and study subjects. Yet this aspect might be critical for drawing conclusions as to what is the putative role of the glyoxalase system present in EVs. A number of women undergoing amniocentesis is lacking as well as the spectrum of indications (just age since the age range is 35-40 or other suspect pathologies such as gestational diabetes?). Speculation on the effect of age and pregnancy pathology should be included in the Discussion section as a possible study limitation. Was the yield of amniotic stem cells and EVs comparable in all samples? What kind of RAGE (ec- or s-RAGE?) would authors expect to detect in EVs?
Author Response
Dear Reviewer,
thank you very much for providing helpful comments and suggestions aimed at improving the manuscript and strengthen the impact of our research work.
Detailed responses addressing point-by-point the issues raised in the individual comments are provided below.
"The MS of Romani R et al. “The glyoxalase system is a novel cargo of amniotic fluid stem 2 cell-derived extracellular vesicles”, presents novel and interesting data. It would be potentially interesting to readers and for the Journal, however, critical details are missing to support the conclusions and interpretation of findings in the present version."
Authors’ response: The authors thank the Reviewer for his/her appreciation on our study and comments/suggestions aimed at improving the manuscript and strengthen the impact of our research work.
"Authors provide very scarce information on the source of biological material and study subjects. Yet this aspect might be critical for drawing conclusions as to what is the putative role of the glyoxalase system present in EVs. A number of women undergoing amniocentesis is lacking as well as the spectrum of indications (just age since the age range is 35-40 or other suspect pathologies such as gestational diabetes?). Speculation on the effect of age and pregnancy pathology should be included in the Discussion section as a possible study limitation."
Authors’ response: As already reported in M&M of the present Communication (“2.4. HASC-derived EVs characterization…….Notably, HASC-P10 and HASC-P100 fractions used in the present study represent aliquots from those used in Mezzasoma et al. [27]…..”), we used residual material of HASC-P10 and HASC-P100 fractions recently used in the study by Mezzasoma et al., Ref. 27. The HASCs from which the P10 and P100 EVs were obtained came from human amniotic fluid collected from different pregnant women over time. These HASCs had already been previously characterized (doi: 10.1111/jcmm.12534 (Ref. 26); doi: 10.1039/c5mb00018a), and turned to be identical, stable for numerous passages (over 200) and did not undergo phenotypic changes during cryopreservation. Hence, since all the HASCs were identical over the other for all the parameters considered in their characterization, including the proteomic profile (doi: 10.1111/jcmm.12534 (Ref. 26), doi: 10.1039/c5mb00018a), only one HASC cell line was considered to recover EVs. A short informative paragraph has been now added in M&M at “2.2. HASCs isolation and culture”. Moreover, we reassure the Reviewer that the women considered had no gestational diabetes or other previous disease and that they were homogeneous for age. When previous diseases were present, women were not included in the study. For the sake of clarity, we have now added a relative informative sentence in M&M.
"Was the yield of amniotic stem cells and EVs comparable in all samples?"
Authors’ response: As above mentioned, we obtained EVs from one HASC cell line.
"What kind of RAGE (ec- or s-RAGE?) would authors expect to detect in EVs?"
Authors’ response: The Ab used for the detection of RAGE expression [anti-RAGE (A-9) mAb, sc-365154] is a mouse monoclonal Ab specific for an epitope mapping between amino acids 23-43 at the N-terminus of RAGE of human origin. This means that it detects a region of the protein common to both full length cell surface or soluble (sRAGE), including the endogenous secretory RAGE (esRAGE) (doi: 10.2119/2007-00087.Koyama). Anyway, we found no RAGE protein expression. We added a short comment on this in Results, paragraph 3.4.
Round 2
Reviewer 2 Report
The authors have done a good job revising the manuscript. I have no further questions.
Reviewer 3 Report
Authors addressed all criticism adequatelly.